# Smuggling on the Nanoscale—Fusogenic Liposomes Enable Efficient RNA-Transfer with Negligible Immune Response In Vitro and In Vivo

**DOI:** 10.3390/pharmaceutics15041210

**Published:** 2023-04-11

**Authors:** Marco Hoffmann, Sven Gerlach, Masanari Takamiya, Samar Tarazi, Nils Hersch, Agnes Csiszár, Ronald Springer, Georg Dreissen, Hanno Scharr, Sepand Rastegar, Tanja Beil, Uwe Strähle, Rudolf Merkel, Bernd Hoffmann

**Affiliations:** 1IBI-2: Mechanobiology, Institute of Biological Information Processing, Forschungszentrum Jülich, 52428 Jülich, Germany; 2Institute of Biological and Chemical Systems-Biological Information Processing (IBCS-BIP), Karlsruhe Institute of Technology (KIT), Postfach 3640, 76021 Karlsruhe, Germany; 3IAS-8: Data Analytics and Machine Learning, Institute for Advanced Simulation, Forschungszentrum Jülich, 52428 Jülich, Germany

**Keywords:** fusogenic liposomes, mammalian cells, cortical tissue, zebrafish, RNA-transfer, lipofection

## Abstract

The efficient and biocompatible transfer of nucleic acids into mammalian cells for research applications or medical purposes is a long-standing, challenging task. Viral transduction is the most efficient transfer system, but often entails high safety levels for research and potential health impairments for patients in medical applications. Lipo- or polyplexes are commonly used transfer systems but result in comparably low transfer efficiencies. Moreover, inflammatory responses caused by cytotoxic side effects were reported for these transfer methods. Often accountable for these effects are various recognition mechanisms for transferred nucleic acids. Using commercially available fusogenic liposomes (Fuse-It-mRNA), we established highly efficient and fully biocompatible transfer of RNA molecules for in vitro as well as in vivo applications. We demonstrated bypassing of endosomal uptake routes and, therefore, of pattern recognition receptors that recognize nucleic acids with high efficiency. This may underlie the observed almost complete abolishment of inflammatory cytokine responses. RNA transfer experiments into zebrafish embryos and adult animals fully confirmed the functional mechanism and the wide range of applications from single cells to organisms.

## 1. Introduction

In recent years, the use of RNA molecules in therapeutic approaches became an increasingly common practice [1,2]. Vaccination strategies also benefit from mRNA technologies [3], as shown for the most promising strategy against the SARS-CoV2 [4,5]. Therefore, the development of efficient and customized transfer reagents and methods became extremely important. In mRNA-based immunization methods, stimulation of the immune system by the transfer mechanism itself has a strengthening effect on the corresponding therapy and is, therefore, a highly beneficial side effect [6,7]. Most methods, however, are intended for efficient transfer of RNA molecules into cells. Examples are therapies targeted against cancer or genetic diseases [8,9]. Here, immune response and an induction in inflammation must be avoided.

During past years, major progress was made to protect nucleic acids from being recognized by the immune systems upon systemic or local application [10], e.g., by use of PEGylated lipids to form so-called stable nucleic acid lipid nanoparticles (SNALPs) [11,12,13]. Nevertheless, the last step of transfer, incorporation into cells, remained highly challenging. This is primarily because these particles are taken up via passive or active, receptor-driven endosomal uptake routes that ultimately transfer RNA complexes into lysosomes for molecular degradation [14]. Along this way, the delivered RNA molecules encounter so-called pattern recognition receptors (PRRs) that are parts of the innate immune system and are enriched in endosomes [15]. PRRs are, for example, responsible for the recognition of foreign nucleic acids after viral or bacterial infection [16,17] and, therefore, recognize almost any type of therapeutic nucleic acid. Upon recognition of foreign nucleic acid molecules, PRRs stimulate especially inflammatory cytokines interleukin 6 (IL-6) and tumor necrosis factor alpha (TNF-α), and ultimately induce cellular apoptosis to prevent the spread of viral and bacterial infections [17,18].

In the central nervous system (CNS), inflammatory responses are prolonged [19]. Therefore, establishing RNA transfer methods that do not trigger inflammatory responses is of special importance for the success of RNA-based therapies of the CNS. Here, one of the most important application fields would be to trigger neurogenesis after brain injuries because in the adult brain of mice, and likely also of humans, neurogenesis is limited to just two compartments, the subependymal/subventricular zone of the telencephalon and the subgranular zone in the hippocampus [20]. In all cases of CNS injury by external force application, reduced amounts of oxygen, blocked or leaking blood vessels, infections or tumors, recovery would massively benefit from activation of post-embryonic neurogenesis. This beneficial function was shown for several long non-coding RNAs which demonstrated their therapeutic potential for brain injury treatment [21]. Further evidence for the potential of this strategy comes from zebrafish, where brain damage or injury are efficiently repaired due to their high regenerative capacity [20,22,23,24,25]. In sum, restoring neurogenesis in the adult mammalian brain is a promising strategy towards therapy of brain injuries [26].

In line with the problems reported for endosomal uptake of therapeutic RNA, first experiments along these lines [24,25,27] pinpointed cytotoxicity and inflammatory response as major challenges. This may have been caused by the use of commercially available lipofection-based carrier systems as detailed described above. Although several delivery systems were tested in the past, most of them completely depended on endocytosis due to lipofection-based carrier systems that ultimately led to impaired biocompatibility and induction in the innate immune response [28].

By using commercial liposomal fusogenic nano-carriers, which immediately fuse with the cellular plasma-membrane [29,30], complexed RNA molecules can be transferred directly into the cytoplasm [31,32]. In contrast to other available lipofection-based transfer systems, fusogenic liposomes bypass the endosomal uptake routes [33]. Therefore, we used fusogenic liposomes to transfer RNA and found highest efficiencies and biocompatibility in in vitro and in vivo applications. Transfer efficiencies go along with minimal inflammatory responses with essentially identical beneficial results in cell culture applications as well as in in vivo treatments, as shown for zebrafish embryos, zebrafish whole brain, and mammalian brain cortical tissue.

## 2. Material and Methods

### 2.1. Cell Culture

Possible biocompatibility effects upon transferring eGFP-mRNA (WOTL19570 L-7601-1000, TriLink) were assessed using the cell lines Chinese hamster ovary K1 (CHO-K1) (ATCC, UK) and pheochromocytoma-12 (PC-12) (ATCC, UK) as well as primary neonatal normal human epidermal keratinocytes (nHEK) (CellSystem, Germany). Freshly isolated primary cortical neurons from E18 old rat embryos were isolated as previously described [34]. Briefly, cortices were isolated from the embryonic brain, treated with trypsin for cell separation, and plated as previously described. All cell types were cultivated corresponding to their specific cell culture medium (see Table 1) at 37 °C in a humidified atmosphere containing 5% CO_2_. Cells were seeded 24 h before treatment either on µ-dish substrates (growth area 3.5 cm^2^, ibidi GmbH, Germany) for confocal microscopy imaging or on glass substrates (growth area 3.5 cm^2^) coated with poly-L-lysine for primary neurons [34] (PLL, Sigma, USA) and on 24 well plates (growth area 1.9 cm^2^, Thermo Scientific, USA) for flow cytometry and qRT-analysis. For all experiments, used cell lines were plated in a density of 50,000 cells/cm^2^ (CHO-K1), 85,000 cells/cm^2^ (PC-12), 20,000 cells/cm^2^ (nHEK), and 170,000 cells/cm^2^ (neurons), respectively.

### 2.2. Ethics Statement for Neuron Isolation

For primary cell isolations from rat, the State Agency for Nature, Environment and Consumer Protection (Landesamt für Natur, Umwelt und Verbraucherschutz North Rhine-Westphalia; number 81-02.04.2019.A396) approved all animal experiments, which were conducted under the national Law for Animal Protection and European regulations and guidelines.

### 2.3. Preparation of Fusion- and Lipofection-Based Reagents

For all experiments, commercially available transfection reagents, namely Fuse-It-mRNA (beniag GmbH, Germany) and Lipofectamine^®^ 2000 (Thermo Scientific, USA), were prepared according to the manufacturers’ instructions. Underlying characterization analysis of Fuse-It-mRNA-based liposomes were performed earlier [31,32] Then, Lipofectamine^®^ 2000 standard protocol was adapted in numbers of transferred eGFP-mRNA molecules equally to Fuse-It-mRNA to comparatively analyze biocompatibility of both transfer mechanisms. To efficiently remove cell-surface bound lipoplexes after transfection reagent incubation, cells were washed after incubation of lipoplexes or fusogenic liposomes three times with 1× PBS or 1× PBS supplemented with Heparin (20 U/mL) at room temperature (RT) [35]. eGFP-mRNA was used throughout all experiments in a concentration of 4 µg/mL for fusion and 2 µg/mL for lipofection. In all cases, a total of 1 µg was transferred per 1.9 cm² substrate. Detailed transfection protocols used in this manuscript are listed in Table 1. For some experiments, the fluorescent compound, as part of Fuse-It-mRNA composition, was tracked according to manufacturer instructions.

### 2.4. Fluorescence Microscopy

Live cell analyses and visualization of expressed fluorescent eGFP signals were performed 24 h after treatment with Fuse-It-mRNA or Lipofectamine^®^ 2000 at 37 °C and 5% CO_2_ using an inverse confocal laser scanning microscope (cLSM 710, Carl Zeiss Jena, Germany). All images showed a representative overview of the center of the substrate and were recorded with an EC “Plan-Neofluar” 10×/0.30 Ph1 objective (Carl Zeiss Jena, Germany). For all experiments, eGFP-fluorescence was excited by a 488 nm argon laser. Emission was detected in a range of 500–550 nm. Throughout all experiments, the microscope settings were kept unchanged to allow for the best data comparability.

### 2.5. Flow Cytometry

Directly after live cell imaging, the flow cytometry (CytoFLEX S Flow Cytometer, Beckmann Coulter) was used to determine eGFP-mRNA expression efficiencies and fluorescence intensities. Briefly, cells were trypsinized (0.05% trypsin-EDTA solution, Sigma Aldrich, USA) and centrifuged at 300× *g*. Without fixation, all cells were analyzed directly after resuspension in 250 µL of the corresponding culturing medium. Due to different morphological cell type characteristics, each cell population was first gated separately on the forward scatter and at least 10,000 cells were analyzed for membrane granularity and cell size. eGFP expression efficiencies and fluorescence intensities were measured by using appropriate filter settings and gatings. 

### 2.6. Live/Dead Stain Assay

To evaluate cell viability by flow cytometry 24 h after transfection the LIVE/DEAD™ Fixable Red stain fluorescence assay (L23102, Invitrogen, USA) was used as described by the manufacture. To measure all dead cells also, cells from the supernatant were included by centrifugation. 

### 2.7. RNA Isolation and cDNA Synthesis for In Vitro and In Vivo Studies

To quantify transferred eGFP-mRNA amounts in vitro and to analyze subsequent immune responses, total RNA was isolated with the RNeasy Plus Mini kit (QIAGEN GmbH, Germany) 24 h post transfection with Fuse-It-mRNA or Lipofectamine^®^ 2000. cDNA synthesis was performed with the QuantiTect Reverse Transcription Kit (QIAGEN GmbH, Germany) after RNA concentration was measured via UV absorption at A_260 nm_ (Nanodrop Products, USA) and diluted to 0.5–1.0 µg with Rnase free water. The isolation of total RNA from zebrafish telencephalon was performed with TRIzol reagent according to manufacturer’s instructions (Invitrogen, Thermo Scientific, USA). The isolated total RNA was dissolved in 10 µL of DEPC-treated water (Ambion, USA) and treated with RQ1 Rnase-Free Dnase (Promega GmbH, Germany). For cDNA synthesis, total RNA from zebrafish telencephalon was used according to the manufacture’s protocol for SuperScript^®^ IV Reverse Transcriptase (Thermo Scientific, USA). 

### 2.8. qRT-PCR Assays for In Vitro and In Vivo Studies

After cDNA synthesis 0.5–1.0 µg of cDNA was diluted 1:1 in RNase free water to characterize cDNA amounts of interest by using TaqMan Assay with specific primers and TaqMan master mix (Thermo Scientific, USA). Used primers are listed in Table 2 for in vitro experiments. For zebrafish in vivo studies, the probes glyceraldehyde-3-phosphate dehydrogenase (GAPDH) zebrafish (Thermo Scientific; Dr03436842_m1 GAPDH), IL-6 zebrafish (Thermo Scientific; Dr03114368_m1 IL1b); and TNF-α zebrafish (Thermo Scientific; Dr03126850 m1 TNF-α) were used. With a StepOne Real-Time PCR System (Thermo Scientific, USA), transferred eGFP-mRNA molecules and expression levels of pro-inflammatory cytokines were quantified for in vitro and in vivo studies. Evaluation was carried out by StepOne Software (version2.0.2).

### 2.9. Inhibition of Endocytosis

For analyzing the effect upon inhibition of endosomal pathways during transfection of both transfer mechanisms, freshly isolated rat embryonic cortical neurons were seeded on PLL coated glass-bottom substrates (growth area 3.5 cm^2^). After cultivating for 9 days, the cells were incubated with 2 mM endosomal inhibitor Methyl-β-cyclodextrin (β-MCD) (Sigma Aldrich, USA) in OptiMEM (Gibco, USA) with 1× PenStrep (Thermo Scientific, USA). Incubation of endocytosis blocking reagents was performed at 37 °C in a humified CO_2_ incubator for 2 h. After incubation, cells were carefully washed twice with warm neurobasal medium before transfection. Appropriate controls to assess cell viability during endocytosis block for untransfected cells were included.

### 2.10. Live Cell Calcium Imaging

Live cell calcium imaging was conducted for analyzing the effect upon both transfer mechanisms regarding functionality and synchronization on primary cortical neuronal network. For that, isolation and seeding of primary neurons was performed as described above. Neurons were cultivated until transfection with eGFP-mRNA for both transfer mechanisms on day 9. Cells were labeled 48 h after transfection with a red fluorescent calcium indicator, Cal-590 AM dextran (AAT Bioquest). Neurons were incubated with 0.5 µM Cal-590 AM in HEPES-buffered HBSS media (0.02 M HEPES, and 0.04% Pluronic acid in HBSS) for 2.5 hand kept for additional 10–30 min in Epatch media (120 mM NaCl, 3 mM KCl, 1 mM MgCl_2_, 10 mM HEPES, 2 mM CaCl_2_) for imaging. Live cell calcium imaging was carried out with an upright microscope (Imager M2, Carl Zeiss, Germany) equipped with a W Plan-Apochromat 20×/1.0 DIC objective (Carl Zeiss Jena, Germany). All images were acquired with an Andor Neo 5.5 sCMos camera at a frame rate of 66 fps for 1 min. Three to four different positions were imaged for each substrate (250–150 cells/field). Each experimental condition was repeated 3–5 times, totaling 1250–450 cells/condition. All images were analyzed using a self-programmed python script, see below. For statistical analysis, 9–20 images per group were analyzed with 250–150 cells per image.

### 2.11. Neuronal Meshwork Synchronization Analysis

Cortical neurons were detected based on the fluorescent signal after Cal590 staining as indicated above. For optimal detection, the first 100 images were averaged and smoothed by a Gaussian filter (width 1 pixel). After determination of an intensity threshold for background foreground separation (Otsu method) and subtraction, all negative values were set to zero. Then, a binary mask was created where all values above zero were defined as cell mask. This mask was post-processed by binary opening. The image that was used for binarization was also used for the watershed algorithm to determine the watershed basins. The watershed basins and the cell mask were multiplied to separate cell clusters. Cell labels with a size less than 20 pixel were discarded. In the next step, the average gray values for each cell label over time were extracted from the images. Each twenty successive time points were averaged to reduce noise. Furthermore, the slowly varying signal for each cell signal was determined by the method of Eilers and Boelens [36] and subtracted from the original signal. This resulted in a flat baseline. Cells with signal amplitudes (maximum minus minimum value) above 1000 gray values were defined as active cells. 

Signal intensity peaks with amplitudes exceeding 1000 gray values were detected using Python’s “sycipy.signal.find_peaks function”. Next, the total number of peaks at each time point was counted. Synchronized time-points were defined as 40% or more events out of all possible peaks. Peak events with a time shift of up to two time points before or after the main peak event were also considered as synchronized.

### 2.12. In Vivo Experiments with Zebrafish

#### 2.12.1. Ethics Statement

Animal husbandry and experimental procedures were performed in accordance with the German animal protection regulations and were approved by the Government of Baden-Württemberg, Regierungspräsidium Karlsruhe, Germany (AZ35-9185.81/G-272/12 and AZ35-9185.81/G-288/18).

#### 2.12.2. Fish and Transgenic Lines

We used the wildtype line ABO (European Zebrafish Resource Centre [EZRC], Karlsruhe). Zebrafish (*Danio rerio*) embryos were maintained at 28.5 °C as described [37,38]. 

#### 2.12.3. Injection into the Adult Zebrafish Brain

For in vivo applications, Lipofectamine^®^ 2000 and specifically adapted, concentrated components of fusogenic liposomes (FLs) were injected into the adult telencephalon as described [25]. In case of FLs, 5× concentrated FLs and 10x concentrated neutralization buffers (according to Fuse-It-mRNA—kindly provided by beniag GmbH) were used to reduce the injected volume. Preparation of FLs was performed with same ratios (*w*/*v*) and buffer conditions according to in vitro FL preparation with 10 µg mRNA to ensure final mRNA concentrations of 0.3 ng/µL. For lipofection-dependent in vivo transfer, Lipofectamine^®^ 2000 was prepared according to the protocol previously described by Ando et al. [39]. Fish were anesthetized with 0.0168% (*w*/*v*) MS-222 and immobilized in the groove of wet tissue papers with the dorsal side up under the dissection microscope. A 30-gauge needle tip was inserted into the cranium at the boundary where the frontals posteriorly meet the parietals [40] to make a window for injection. Approximately 5 µL of lipofection or FL samples were pipetted with a 20 µL microloader tip (Eppendorf, Germany) and filled into a glass capillary (TW100F-4, 4″, w/Fill, 1.0 mm; World Precision Instruments, Germany) pulled with a Flaming/Brown micropipette puller (model P-97, Sutter Instrument Co., USA). Microinjection was performed by Femtojet microinjector (Eppendorf, Germany). After injection, fish were recovered in fish water supplemented with 10 units/mL of penicillin and 10 µg/mL streptomycin (Gibco, Thermo Scientific, USA).

#### 2.12.4. Injection into the Zebrafish Embryo

The eggs spawned by wildtype zebrafish were raised to the dome, or 30%-epiboly stage (4.5 hpf) in E3 medium. Lipoplexes of Lipofectamine 2000 or FLs were injected into the yolk of the embryo at the site close to the blastoderm by using the setups described for the adult fish brain injection.

#### 2.12.5. Immunohistochemistry of Fish Brains

Adult zebrafish were killed with 0.04% (*w*/*v*) MS-222 and the brain was excised under a dissection microscope by using forceps and suspended briefly in PBS 24 h after injections. The dissected brain was fixed in 4% (*w*/*v*) paraformaldehyde/PBS at 4 °C overnight. The fixed brain was washed twice with PBS and post-fixed in 100% methanol at −20 °C overnight. The brain was then rehydrated through a descending methanol series (75%, 50%, 25%, and 0% methanol in PBS) and embedded in 2% (*w*/*v*) low-melting agarose for sectioning. Then, 50-µm thick transverse brain sections were made by a vibratome (VT1000S, Leica Biosystems, Nussloch, Germany). Brain sections were treated with blocking buffer (0.2% (*w*/*v*) bovine serum albumin, 1% (*v*/*v*) dimethyl sulfoxide, 0.1% (*v*/*v*) Tween-20 in PBS) and processed for immunohistochemistry to detect eGFP by using chicken anti-green fluorescent protein (GFP) (1:1000, Aves Labs, USA) as a primary antibody. Anti-chicken Alexa 488-conjugated antibodies (1:1000, Invitrogen, USA) were used as secondary antibody. Processed slices were mounted in aqua polymount (PolyScience, USA) for confocal imaging.

#### 2.12.6. Fluorescence Microscopy of Zebrafish

Zebrafish embryos were mounted in 0.5% low-melting agarose (Carl Roth, Karlsruhe, Germany) in E3 medium [37,38] supplemented with 0.0168% (*w*/*v*) MS-222 (Sigma Aldrich, USA) as an anesthetic. Imaging was performed with an upright confocal microscope (TCS SP5, Leica Microsystems, Germany) with HCX PL APO 20×/0.7 and HCX APO 20×/0.5 objectives for the adult brain and the embryo, respectively. For all experiments, eGFP-fluorescence was excited by a 488 nm argon laser and the fluorescent marker dye compound of Fuse-It-mRNA by a 633 nm helium-neon laser. Emission was detected in the range of 500–550 nm (eGFP signal) and 650–700 nm (fluorescent marker dye of Fuse-It-mRNA). Throughout all experiments, the microscope settings were kept unchanged to allow for the best data comparability.

#### 2.12.7. Quantification of eGFP-Positive Cells in Zebrafish Telencephalon-Slices

eGFP-positive cells in telencephalon slices were identified using an in house CoverageTracker routine (Matlab R2018a) based on co-localization analyses with DAPI-stained nuclei. First, the images of the channels with the DAPI-stained cell nuclei and with the eGFP signal were separated in single image stacks. Local maxima were calculated for each image stack pixel by pixel (in z-direction) and maxima were saved in a single image. These images were smoothed by Gaussian filtering, normalized, and locally contrast-enhanced (CLAHE = contrast-limited adaptive histogram equalization) [41]. The local adaptive threshold method according to Bradley was used to generate binary images [42]. Additionally, the excentricity of objects in the eGFP signal image was used as a second threshold value to reduce wrong signal ranges (epsilon between 0 for circles and 1 for lines). The degree of coverage was the quotient of the sum of all areas with an eGFP signal to the sum of all DAPI-stained cell nuclei areas.

### 2.13. Statistical Analysis

All statistical analyses were performed using one way ANOVA (analysis of variance) with post hoc Tukey HSD (honestly significant difference) Test. *p*-values: not significant (n.s.): *p >* 0.05, *: *p* ≤ 0.05, **: *p* ≤ 0.01, ***: *p* ≤ 0.001.

## 3. Results

### 3.1. Quantity of Transferred Nucleic Acids and Transfer Mechanism Determine Biocompatibility in Neuronal Cells

In order to optimize RNA-transfer into neuronal cells with respect to efficiency and biocompatibility for subsequent in vivo applications, we compared classical lipofection systems with fusion-based transfer reagents. The transfer of eGFP-mRNA into the neuronal-like cell line PC-12 by lipofection and fusion standard protocols resulted in reproducible transfer efficiencies and fluorescence intensities that depended on the transfer method (Figure 1a,b). Here, the lipofection-based transfer resulted in approximately 25% more eGFP-positive cells than fusion and a more than 25-fold higher fluorescence intensity was detected (Figure 1e). The amount of transferred mRNA molecules was 15-fold increased based on qRT-PCR analyses (Figure 1f). However, this standard lipofection protocol also resulted in more than 20% higher numbers of dead cells (Figure 1e) and approximately 15-fold higher expression of the pro-inflammatory cytokine IL-6 than the standard fusion protocol (Figure 1f). 

The results argued for enhanced cellular inflammation response due to high concentrations of transferred nucleic acids and prolonged incubation for lipofection. We, therefore, tried for best comparability of both methods to reduce cytotoxicity by shortened incubation periods of just 40 min upon lipofection and, in addition, the removal of remaining cell-attached lipoplexes by washing with PBS-heparin for both transfer methods. As a result, the transfection efficiency for lipofection dropped by more than 50%, while it remained unaffected upon fusion (Figure 1c,d). With these adaptations, both fusion and lipofection resulted in comparable eGFP-mRNA transfer and fluorescence intensities (Figure 1e,f). While this improved lipofection protocol resulted in clearly reduced IL-6 expression as well as induced cell death, these values were still significantly higher than for fusion (2.5-fold increased cell death and approximately 4-fold elevated IL-6 expression). Interestingly, the removal of surface attached fusogenic liposomes by heparin washing had no effect on mRNA transfer, GFP fluorescence intensity, and IL-6 expression. The only significant effect of heparin washing was a further reduction in the number of dead cells (Figure 1e,f). 

We found very similar results with pronounced correlations between transferred mRNA amount and fluorescence intensity for other cell types (Appendix A). Interestingly, even the mild lipofection protocol resulted in massive apoptosis with, e.g., more than 80% dead cells for very sensitive primary nHEK cells within 24 h, while fusion did not affect cell viability at all. For lipofection, cell death went along with strongly enhanced IL-6 (around 400-fold) as well as TNF-α (150-fold) expression levels compared to endosomal-independent fusogenic eGFP-mRNA transfer (Appendix A).

### 3.2. The Reduced Influence of Endosomal Uptake Mechanisms for Fusion-Based RNA Transfer Allows High Biocompatibility and Low Cytokine Responses

Cell lines can come along with artificial cell behavior. For this reason, we also studied primary rat cortical neurons with respect to cytokine induction after lipofection or fusion in the absence and presence of blocked endocytosis. For lipofection, we found a pronounced transfer reduction upon blocked endocytosis. Here, qRT-PCR-analyses proved a strongly reduced amount of transferred eGFP-mRNA molecules upon block of endocytosis that correlated with impaired eGFP signals (Figure 2d). The same analyses after fusion showed just minor effects on both. 

The quantification of cell viability and cytokine responses in primary cortical neurons further supported the strong link between endocytic transfer routes and reduced biocompatibility. Here, endocytosis-dependent lipofection resulted in a two-fold increased number of dead cells (Figure 2e) and an elevated level of IL-6 mRNA by a factor of two compared to fusion-based transfer (Figure 2g). Surprisingly, without blocking endocytosis, the cytokine response was significantly enhanced by endocytic uptake, even though quantities of eGFP-positive cells (Figure 2a,c), as well as fluorescence intensities (Figure 2e), showed no significant differences between lipofection and fusion. Upon blocking endocytosis, fluorescence intensities and transferred mRNA amounts dropped for lipofection, while they remained largely unaffected for fusion-based transfer (Figure 2f,h). Interestingly, while unaffected transfer rates for fusion went along with constitutively low IL-6 levels and low numbers of dead cells, blocking endocytosis only reduced IL-6 expression for lipofection-treated cells to levels shown for fusion but the number of dead cells increased even further by a factor of two (Figure 2f,h). 

### 3.3. Endocytosis-Dependent RNA Transfer Blocks Functionality of Neuronal Networks 

For most sensitive biocompatibility assays on neuronal level, functional networks of synchronized primary cortical neurons were formed with more than 90% of all cells communicating with each other (Figure 3; see also control sample in Appendix A). Visualization of synchronous membrane potential depolarization (Calcium (Ca^2+^) inflow) as well as addition of endosomal blocker did not affect functionality (Figure 3a). The endosome-independent nucleic acid transfer by fusion 48 h before analysis slightly reduced the overall number of synchronized cells but did not influence synchrony or the number of synchronized membrane potentials over time. Interestingly, blocking endocytosis upon fusion-dependent mRNA transfer had no effect on synchronized network functionality but slightly increased number of synchronized events (Figure 3b). In contrast, endosome-dependent lipofection fully blocked detectable depolarization events and, therefore, no synchronous cell communication could be detected (Figure 3c). While blocking endocytosis for the complete incubation time after lipofection killed neurons, during lipofection, it reduced cytotoxicity and enabled some single neurons to regain their ability and form spontaneous Ca^2+^ events. In addition, the number of cells were clearly reduced after lipofection (83 cells s.d. 30 cells per field of view). This number significantly increased upon blocking endocytosis (167 cells s.d. 100 cells). These data confirm strongly enhanced biocompatibility upon direct nucleic acid transfer into the cytoplasm of neurons. 

### 3.4. Inflammatory Responses Are Also Initiated In Vivo by Endosomal Transfer Mechanisms of Nucleic Acids

We next assessed whether the transfer of mRNA occurs efficiently in tissue of intact animals. We transferred eGFP-mRNA into cells of the telencephalon of adult zebrafish by lipofection and fusion by injection into the ventricle of the telencephalon. In all cases, the fish remained vital without any sign of morphological or neurological impairments. A total of 24 h after transfer, the fusion-based mRNA transfer resulted in high GFP expression in cells of the subventricular domain and widespread and homogeneously distributed lower expression in both telencephalic hemispheres (Figure 4a, and Appendix A). DAPI co-localization analyses on 10 independent telencephalon slices proved transfer efficiencies of eGFP-mRNA into 35% (s.d. 9.4%) of all cells. In contrast, in lipofected animals, scattered eGFP-positive cells were exclusively present in close proximity to the ventricle. These cells showed higher eGFP intensities with extensions visible for some cells, likely the processes of radial glial cells. Consequently, overall transfer efficiencies were low (5.3% (s.d. 4.1%) eGFP-positive cells) (Figure 4a).

Strongly increased IL-6 as well as TNF-α expression was detected in lipofected brains, suggesting a strong inflammatory response by endosome-dependent transfer also on an in vivo level (Figure 4b). The levels were much higher than for fusion-treated brain tissue. The latter showed just marginally increased inflammatory responses compared to the negative control (PBS injection only). Remarkably, IL-6 and TNF-α expression levels in lipofected brains were even higher than those found after induction with injected lipopolysaccharide (LPS), used as positive control. These data showed that fusion-mediated mRNA transfer leads to a much more uniform expression also deep in the tissue of a living organism without inducing an inflammatory response in comparison to lipofection. 

### 3.5. Endosome-Independent Transfer Mechanisms Are Highly Efficient without Detectable Teratological Effects in Early Embryogenesis

Finally, fusion- and endosome-dependent mRNA-transfer systems were analyzed during the early stages of embryogenesis to verify potentially teratological effects in early embryogenesis. The efficiency of direct transfer of mRNA molecules into the blastomere by microinjection is empirically known to drop sharply after the one-cell stage. Therefore, we selected the later stages, dome or 30%-epiboly (4.5 h post fertilization (hpf)), to examine the efficiency of our transfer system. To this end, zebrafish embryos were injected with lipoplexes of fusogenic or endosome-dependent lipofection into the yolk at the site close to the blastoderm and allowed to develop further. No visible defects in embryogenesis for either group of injected embryos were detected at 24 hpf. However, the transfer of endosome-dependent lipofection failed to induce expression in differentiated cells of the embryo. Instead, strong expression of GFP protein was only detectable in the yolk itself, suggesting that the mRNA transfer by endosome-dependent lipofection during the midblastula stage was limited to the yolk syncytial layer, an extraembryonic structure formed at 3 hpf [43]. This expression pattern did not change after incubation for an additional 24 h (Figure 5). For fusogenic liposomes, however, a broad spectrum of GFP expressing cells could be identified already 24 h after mRNA transfer. Expressing cells were essentially embryonic structures including cells of the neural tube, sensory cells, endothelial cells, and connective tissue. Typically, fusogenic liposome-induced protein signals were more prominent in the cells of the embryo and less in the extraembryonic structure such as the yolk, which argues for a highly efficient and long distance RNA transfer from the yolk into blastomeres.

## 4. Discussion

Efficient and biocompatible nucleic acid transfer methods are essential for many biomedical approaches [44,45,46,47]. However, the use of nucleic acids goes along with challenges due to a large number of conserved extracellular and intracellular mechanisms to protect the organism from non-endogenous nucleic acid information [17,48,49,50]. While bacterial DNA is a potent immune stimulant by virtue of its CpG motifs [51], free mammalian DNA, which is ordinarily inactive, can acquire activity by associating with nuclear, cytoplasmic, and serum proteins which promote its uptake into cells to stimulate internal DNA sensors, including Toll-like receptor (TLR) 9 [52,53]. Another format to assess an inflammatory activation by DNA involves transfection reagent dependent DNA complexation [54] to promote uptake into cells. Importantly, the stimulation of cells by transfected lipoplex-DNA does not require CpG motifs and appears to occur with essentially any DNA due to complexation with positively charged lipids or polymers [55,56] that facilitate intracellular delivery through electrostatic interactions with the membrane of the target cell. Although depending on different receptors, RNA can also effectively stimulate the innate immune response based on endosomal and cytoplasmic receptors [57,58,59,60,61]. As for DNA, delivery vehicles that enable RNA uptake have major effects on inflammatory responses [32,55]. A common feature of such systems is that they are taken up by endocytosis and concentrated in the endosomal compartment before releasing nucleic acids into the cytoplasm. Immune stimulation by RNA may be potentiated by delivery vehicles because of a number of factors, including more efficient uptake into intracellular compartments and protection of RNA from nuclease degradation [62]. RNA encapsulated in SNALPs or complexed with polyethyleneimine exemplarily induce a response that is dominated by IFN-secretion, while RNA complexed with lipofectamine and polylysine form larger, more heterogeneous RNA complexes, and tend to elicit a predominantly inflammatory cytokine response [63]. 

Our results highlight that cationic liposomes can induce cytotoxic effects even without endocytotic uptake. Upon blocking endocytosis, the uptake of lipoplexes was largely prevented to result in a reduced cytokine-release, but the cell viability was still significantly impaired. This effect may be induced by the accumulation of lipoplexes on the plasma membrane, which negatively affects metabolic processes [64], signal transduction [65], and consequently, cell viability [66]. Effects on viability were not observed upon use of fusogenic liposomes, which immediately fuse with the plasma membrane to transfer the mRNA freely available into the cytoplasm. The composition and preparation of the fusogenic liposomes used here was first described by Csiszár et al. [29]. By the development of an appropriate neutralization buffer, which allows an adequate complexation of the nucleic acid and subsequent interaction with FLs without affecting the fusogenic properties of the FLs, these liposomes could be used the first time for the transfer of mRNA [32]. A precise characterization of the functional composition and phase transition of FLs was described by Kolasinac et al. [67]. The characteristics of FLs used for mRNA transfer were described in complex size and surface charge by Hoffmann et al. [31]. They demonstrated that FLs transferred twice as much mRNA within 10 min as classical lipoplexes within 8 h [31]. However, at this moment, we cannot exclude that differences between endocytotic and fusogenic liposomes are also caused, at least in parts, by different mRNA encapsulation efficiencies, although overall transferred mRNA amounts were comparable after adaptation of the lipofection protocol. 

With our experiments, we could show that the transfer route has a crucial influence on the recognition and subsequent processing of the mRNA in treated cells. This may ultimately have a decisive impact on the success of any in vivo application. In view of this, new therapeutic applications may be optimized. The efficient recognition by receptors such as RIG-1, melanoma differentiation-associated protein 5 (MDA5) as well as lipophosphoglycan 5 (LPG5) [17,59,68,69] leads to the subsequent activation of corresponding inflammatory signaling cascades in response to the transfer of dsRNA molecules [60,70]. The direct increase in corresponding cytokines shows the efficiency of the innate immune system for the detection of foreign double-stranded nucleic acids and immune defense. No effects of comparable strength were found upon mRNA transfer with transfer reagents tested here. These comparably minor effects are, of course, favored by the significantly less recognized and immunogenic single-stranded mRNA. However, the reduced expression of inflammatory cytokines, such as TNF-a and IL6, after treatment with fusogenic liposomes in vitro and in vivo may suggest a reduced immune response after treatment with this formulation. The underlying reasons for such enhanced biocompatibility need to be shown in further studies. 

Although lipoplexes with positively charged lipids are used in classical lipofection reagents as well as in fusion systems [67,71] with well described effects on cell viability [64,72], the fate of such lipids after transfer could play an important role. Studies showed that components of fusogenic liposomes remain in the plasma membrane after fusion. Fusogenic liposomes seem to release their contents into the interior of the cell in a mainly uncomplexed manner [33,73]. It is, therefore, likely that positively charged complexation reagents remain rather inactive when primarily present in the plasma membrane without directly entering inner cellular compartments. Such an effect would not only have important implications for neuronal manipulation as performed here, but also for many other medical application fields. Although the use of modified nucleotides such as pseudo uridine or 5′-methyl cytidine for the generation of in vitro transcribed mRNA [74] already significantly reduced endosome-dependent recognition of such mRNA by TLR-3, 7, and 8 [75], skipping such potential detection by use of fusion would further significantly increase the safety of mRNA-based therapies.

Especially for the central nervous system, inflammatory signals represent the basis of highly complex, still only partially understood regulatory pathways. These inflammatory signals, such as the molecules IL-6 or TNF-α also analyzed here, are released by astrocytes and glial cells and can exert both neuroprotective as well as neurodegenerative functions [76]. Typically, low levels of inflammatory signals are associated with a protective function, whereas increased levels have a degenerative effect [76]. Ambitious therapeutic approaches aim to treat neurodegeneration through induced neurogenesis based on the neural stem cells present in mammals [77] and neuronal regeneration [78]. Unfortunately, endosome-dependent lipofection induces a strong inflammatory response. Such inflammation can, at least in zebrafish, be a trigger of adult neurogenesis in the telencephalon [79] and, thus, represents a severe complication of experimental and therapeutic approaches in the brain. 

Because of their special ability to induce neurogenesis efficiently throughout their life cycle, zebrafish represent an indispensable animal system for understanding and developing such therapeutic approaches. A key issue is to induce expression of regulatory molecules in the stem cells lining the ventricular zone to result in differentiating new-borne neurons that migrate into deeper layers of the parenchyma. Although the depth of penetration can be improved by dendrimers [80,81], current lipofection protocols are limited regarding these characteristics [25]. Fusion-based RNA transfer may, thus, allow also to manipulate new-borne neurons in their direction of differentiation or to tune the function of neurons in intact neuronal circuits. It currently remains an open question how the fusion-mediating transfer can reach the deeper layers of the parenchyma. Furthermore, the detection of FLs even in blood vessels as shown in Appendix A might also have important implications for endothelial modification and even blood–brain barrier trafficking of later therapeutic applications. The extraordinary penetration of tissues was also observed in the embryo, where injection into the yolk cell lead to widespread expression in differentiated cells of the embryo proper. In contrast, endosome-dependent expression remained localized in the injected yolk cell without leading to any detectable expression in the embryo. 

The transferability of data on genetic dissection of neural circuits for adult neurogenesis and neuronal regeneration [82] as well as for drug discovery [83] was impressively demonstrated many times. However, to take the next step towards the development of efficient RNA-based therapies, it is crucial to ensure high transfer rates into CNS cells without inducing inflammatory signals by the transfer reagents. Both basic requirements seem to be available with fusion systems for further analysis and development.

## 5. Conclusions

Transfection reagents for mRNA molecules are mainly based on two different transfer mechanisms into animal cells. In a comparison of endosomal-dependent and fusion-driven commercial transfection systems, both mechanisms showed efficient transfer and good expression levels in cell culture. However, comparative biocompatibility analyses showed a clear advantage of fusion-driven mRNA transfer with low cytokine expression and unaffected cell function. By confirming high transfer efficiency and minimal cytotoxic effects for fusion-driven mRNA transfer in both embryonic and adult animals, our data implied a high application potential of fusogenic liposomes for long-term cell culture analyses as well as in vivo experiments and possible therapeutic approaches.

## Figures and Tables

**Figure 1 pharmaceutics-15-01210-f001:**
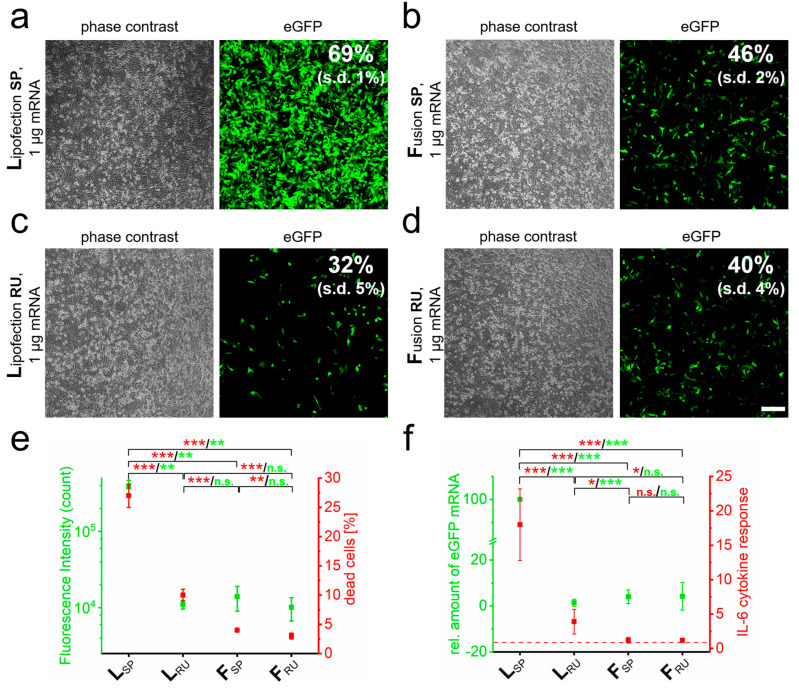
RNA quantity and transfer method influence biocompatibility and immune response. eGFP-mRNA was transferred into PC-12 cells by lipofection (**L**) (**a**) and fusion (**F**) (**b**) based standard protocols (**SP**) and analyzed 24 h after transfer by phase contrast and fluorescence microscopy. Transfection efficiency was determined by flow cytometry. Identical analyses with reduced uptake (**RU**) conditions were performed for lipofection (**c**) and fusion (**d**). Analyses of cells as indicated in a to d for cell death and fluorescence intensity are shown in (**e**). Cells were additionally harvested for mRNA isolation and characterized for IL-6 and eGFP by qRT-PCR (**f**). eGFP-mRNA was used in a concentration of 4 µg/mL for fusion and 2 µg/mL for lipofection. In all cases, a total of 1 µg was transferred per substrate. n = At least three independent experiments were used for each analysis. *p*-values: not significant (n.s.): *p >* 0.05, *: *p* ≤ 0.05, **: *p* ≤ 0.01, ***: *p* ≤ 0.001.

**Figure 2 pharmaceutics-15-01210-f002:**
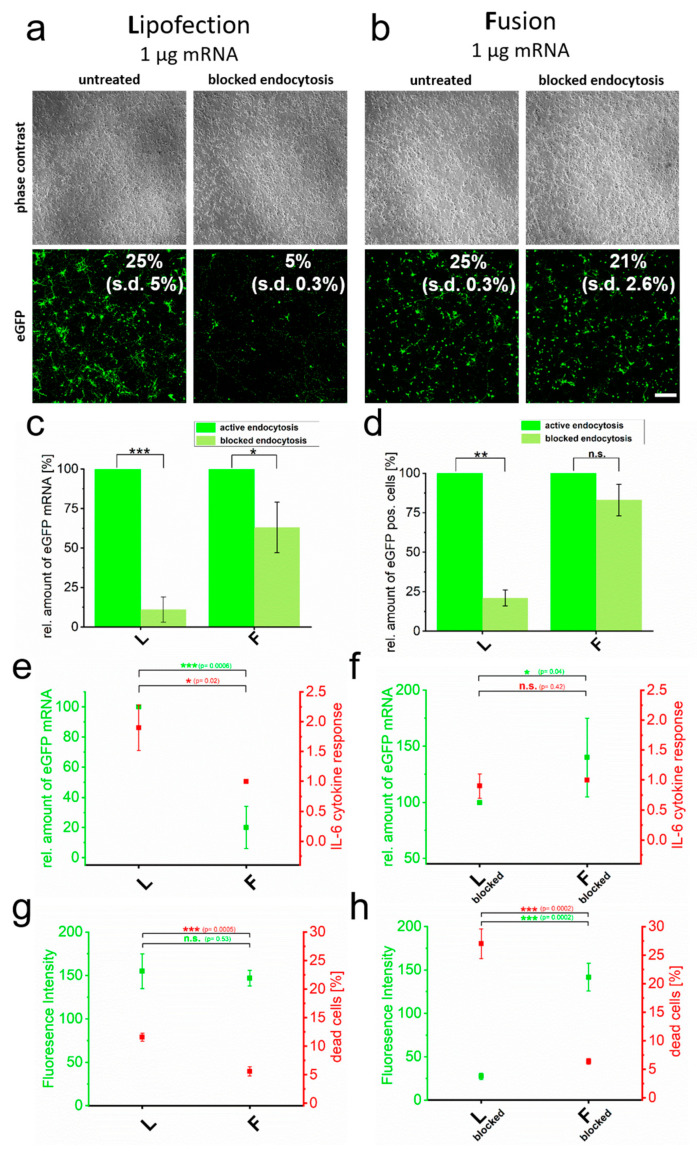
Cytotoxicity induced by RNA transfer is dependent on endocytic uptake. eGFP encoding mRNA was transferred into primary cortical neurons using endocytosis-dependent (lipofection, (**a**)) and -independent (fusion, (**b**)) transfection reagents. Transfer was performed in the absence and presence of endosome inhibitor β-MCD and cells were analyzed 24 h after treatment. Transfection efficiencies are indicated. Identically treated cells were analyzed for eGFP intensity by flow cytometry (**c**), relative amount of transferred eGFP mRNA (normalized to active endocytotic cells (**c**,**d**) and to the respective highest values of each measurement in (**e**,**f**)) (**d**), IL-6 expression (**e**,**f**) and number of dead cells (**g**,**h**). For better comparison, IL-6 values are directly compared to transferred eGFP mRNA levels (**e**,**d**). Percentage of dead cells is given in comparison to fluorescence intensity (**g**,**h**). L = lipofection, F = fusion. eGFP-mRNA was used in a concentration of 4 µg/mL for fusion and 2 µg/mL for lipofection. In all cases, a total of 1 µg was transferred per substrate. Scale bar: 200 µm, *n* = three independent experiments for each analysis. *p*-values: not significant (n.s.): *p >* 0.05, *: *p* ≤ 0.05, **: *p* ≤ 0.01, ***: *p* ≤ 0.001.

**Figure 3 pharmaceutics-15-01210-f003:**
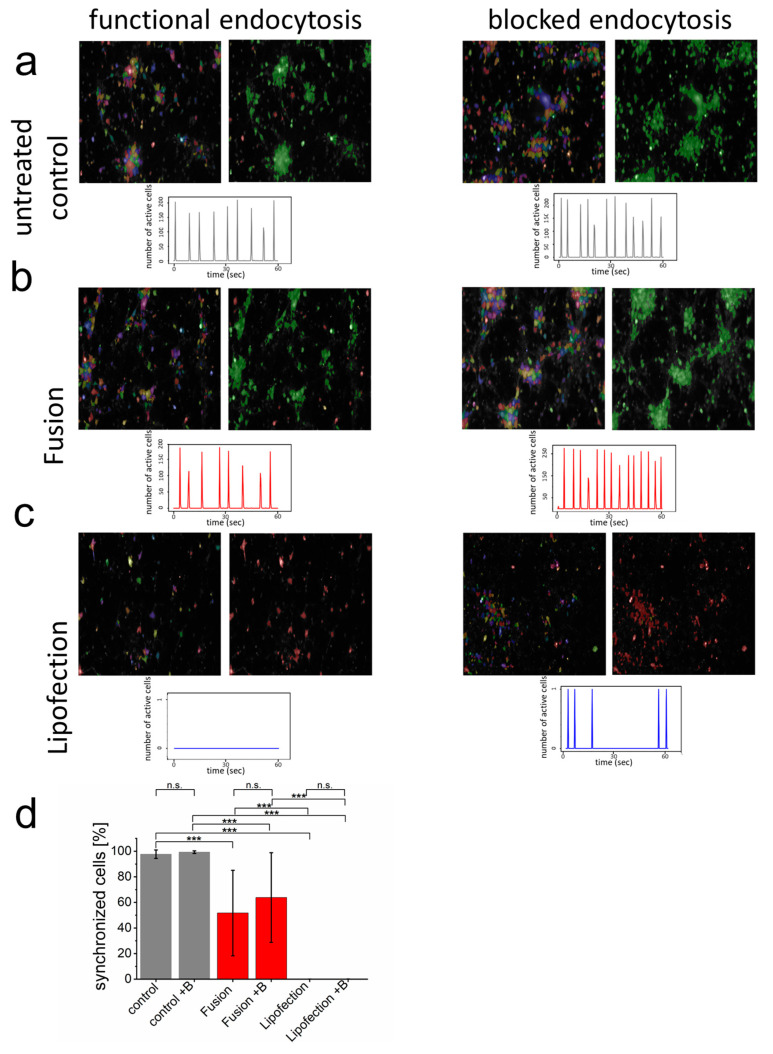
Endocytosis of nucleic acids impairs inter-neuronal network communication. (**a**–**c**) After network formation, primary cortical neurons were analyzed for functional and synchronized Ca^2+^ signaling (left = all identified cells, right = synchronized cells in green and inactive cells in red) in the absence and presence of endosome inhibitor β-MCD. Inhibitor was added to the culture medium two hours before cells were transfected with GFP-mRNA by endosomal-independent ((**b**), fusion) or endosomal-dependent ((**c**), lipofection) transfer mechanisms and compared to the untransfected control (**a**). Statistical evaluation of all experiments including s.d. (**d**) Here, +B: addition of endosomal inhibitor β-MCD. eGFP-mRNA was used in a concentration of 4 µg/mL for fusion and 2 µg/mL for lipofection. In all cases, a total of 1 µg was transferred per substrate. Scale bar = 200 µm. *n* = three independent experiments with at least 9 independently formed neuronal networks. *p*-values: not significant (n.s.): *p >* 0.05, *: *p* ≤ 0.05, ***: *p* ≤ 0.001.

**Figure 4 pharmaceutics-15-01210-f004:**
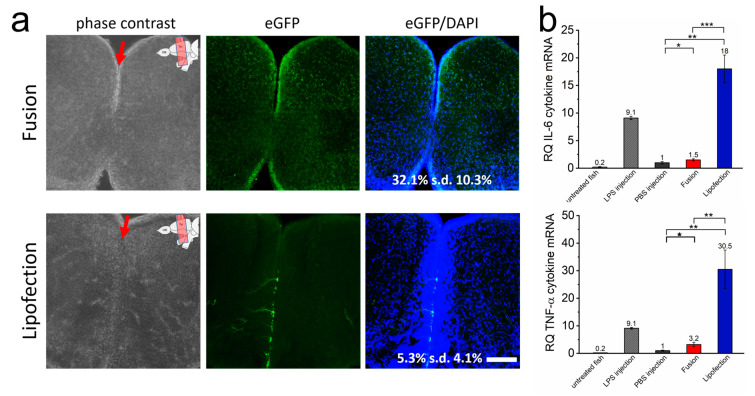
Endosome-independent transfer allows efficient transfection rates with low cytokine response in vivo. (**a**) The brain of adult zebrafish was injected with GFP-mRNA containing fusogenic liposomes (top) or dependent on endosomal uptake (lipofection, bottom). Injection into the ventricle is indicated by a red arrow, upper right of the image shows ventricular zone of zebrafish brain. A total of 24 h after nucleic acid transfer, whole brains were isolated and brain sections were stained for GFP (green) and nuclei (blue). Co-stained nuclei were counted as transfected cells and are given as percentage of all cells. The whole RNA of identically treated brains was isolated 24 h after nucleic acid transfer and tested for IL-6 and TNF-α expression (**b**). LPS was used as positive control, PBS as negative control. eGFP-mRNA was used in a concentration of 0.3 ng/µL (corresponds to 1.5 ng per injection). Scale bar = 200 µm. Each condition was analyzed on at least 5 independent injections. *p*-values: not significant (n.s.): *p >* 0.05, *: *p* ≤ 0.05, **: *p* ≤ 0.01, ***: *p* ≤ 0.001.

**Figure 5 pharmaceutics-15-01210-f005:**
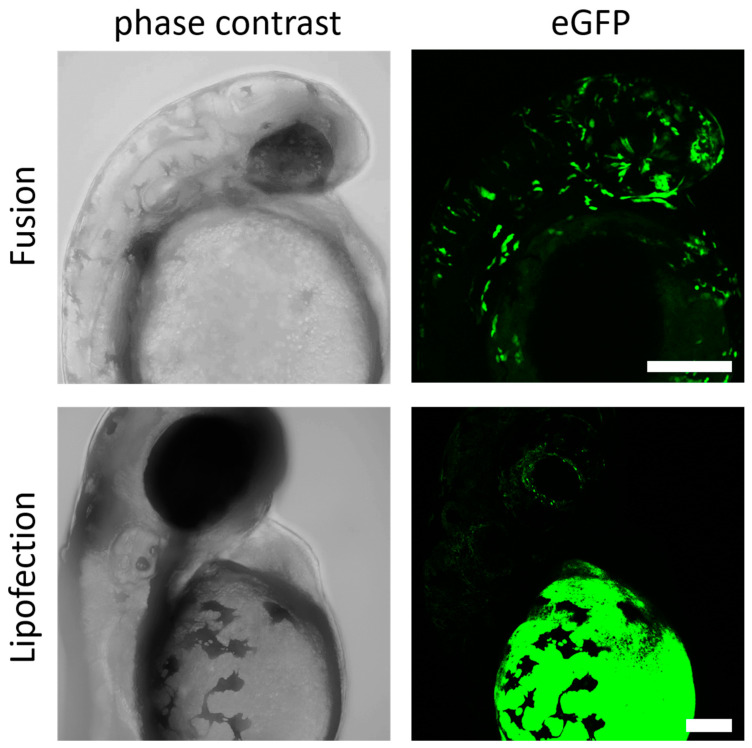
Fusogenic liposomes mediate efficient expression in zebrafish embryos. GFP-mRNA was injected into the zebrafish yolk at 4.5 hpf by endosome-independent (fusion) and endosome-dependent (lipofection) mechanisms. Embryos were analyzed for GFP expression 24 h (fusion) and 48 h (lipofection) after injection. eGFP-mRNA was used in a concentration of 0.3 ng/µL (corresponds to 1.5 ng per injection). Scale bar = 200 µm.

**Table 1 pharmaceutics-15-01210-t001:** Cell culture medium for assessed cell lines and applied transfection protocols. Overview of all media and supplements and transfection protocols applied for each assessed cell line are provided. Amounts are given in (*v*/*v*).

	CHOK1	nHEK	Neuronal-Like PC-12	Primary Embryonic Cortical Neurons
Medium	DMEM F12 Glutmax (Gibco, USA), 1× PenStrep (Gibco, USA), 10% FBS (Superior, Biochrom)	Dermalife K (CellSystems, Germany) without EGF	RPMI 1640 (Gibco, USA), 1× PenStrep (Gibco, USA), 10% FBS (Superior, Biochrom)	50 mL Gibco Neurobasal Medium + 1 mL B-27 Supplement (Thermo Scientific, USA), 2.5 mg Gentamicin Reagent Solution (Sigma, USA), 125 µL GlutaMax (Thermo Scientific, USA)
Lipofectamine^®^ 2000 standard protocol (**LSP ^1^**)	2 µL reagent in 25 µL OptiMEM + 1 µg eGFP mRNA in 25 µL OptiMEM/complexed mRNA + reagent (50 µL) add 500 µL media/incubation over night			
Lipofectamine^®^ 2000 reduced uptake protocol (**RU ^2^**) by reduced incubation time (**LRT ^3^**)	Lipoplex prepared as in L SP add 500 µL OptiMEM/incubation for 40 min (min)			/
Lipofectamine^®^ 2000 reduced uptake protocol (**RU ^2^**) by LRT and PBS washing (**LRT + PBS ^4^**)	Lipoplex prepared as in L SP add 500 µL OptiMEM/incubation for 40 min/wash cells 3× with 1× PBS, pH 7.2, room temperature (RT)			/
Lipofectamine^®^ 2000 reduced uptake protocol (**RU ^2^**) by LRT and PBS-H washing (**LRT + PBSH ^5^**)	Lipoplex prepared as in L SP add 500 µL OptiMEM/incubation for 40 min/wash cells 3× with 1× PBS-Heparin (20 U/mL Heparin-Natrium-25.000 (i.v./s.c.) ratiopharm GmbH), pH 7.2, RT			/
Fuse-It-mRNA standard protocol (**F SP ^6^**)	2 µL neutralization buffer (NB) + 1 µg eGFP mRNA 10 min RT/add 2.5 µL fusogenic solution (FS)			
	10 min fusion in 250 µL PBS	8 min fusion in 250 µL PBS	10 min fusion in 250 µL PBS	10 min fusion in 250 µL PBS
Fuse-It-mRNA reduced uptake protocol **(RU ^2^**) by PBS washing after fusion (**F SP + PBS ^7^**)	after incubation at 37 °C washed 3× with 1× PBS, pH 7.2, RT			/
Fuse-It-mRNA reduced uptake protocol (**RU ^2^**) by PBS-H washing after fusion (**FSP + PBSH ^8^**)	after incubation at 37 °C washed 3× with 1× PBS-Heparin (20 U/mL Heparin-Natrium-25.000 (i.v./s.c.) ratiopharm GmbH), pH 7.2, RT			

Abbreviations of Table 1: ^1^ LSP: Lipofectamine^®^ 2000 standard protocol; ^2^ RU: reduced uptake protocol; ^3^ LRT: Lipofectamine^®^ 2000 reduced incubation time protocol; ^4^ LBS: Lipofectamine^®^ 2000 reduced incubation time and PBS washing protocol; ^5^ LRT + PBSH: Lipofectamine^®^ 2000 reduced incubation time and PBS supplemented with Heparin washing protocol; ^6^ FSP:Fuse-It-mRNA standard protocol; ^7^ FSP + PBS: Fuse-It-mRNA standard and PBS washing protocol; ^8^ FSP + PBSH: Fuse-It-mRNA standard and PBS supplemented with Heparin washing protocol.

**Table 2 pharmaceutics-15-01210-t002:** qRT-PCR Primers used for all assessed and treated cell lines. Overview of all used qRT-PCR Primers for in vitro and in vivo analysis are listed below.

	CHOK1	nHEK	Neuronal-Like PC-12	Primary Embryonic Cortical Neurons
endogenous control	GAPDH chinese hamster (Thermo Scientific; Cg04424039-gh GAPDH FAM)	GAPDH homo sapiens (Thermo Scientific; HS02786624-g1 GAPDH)	GAPDH rattus norvegius (Thermo Scientific; Rn01775763_g1)	
eGFP	eGFP mr Enhance (Thermo Scientific; Mr04097229_mr eGFP)			
IL-6	/	IL-6 Fam homo sapiens (Thermo Scientific; HS00174131_m1 IL-6)	IL-6 Fam rattus norvegius (Thermo Scientific; Rn01410330_m1)	
TNF-α	/	TNF-α homo sapiens (Thermo Scientific; HS00174128_m1 TNF-α)	/	/

## Data Availability

The data of this study are available on request from the corresponding author.

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
