# Peer review of "Smuggling on the Nanoscale—Fusogenic Liposomes Enable Efficient RNA-Transfer with Negligible Immune Response In Vitro and In Vivo"

_pharmaceutics, 2023, doi:10.3390/pharmaceutics15041210_

Round 1

Reviewer 1 Report

The manuscript entitled “Smuggling on the nanoscale-Fusogenic lipsosomes enable efficient RNA-transfer with negligible immune response in vitro and in vivo” aims to compare in culture cells, primary cells and in vivo in zebrafish, the transfection efficiency and inflammatory responses induced by two commercial mRNA-transfectants, that differ in the entry mechanism in the cells: fusion with cell membrane of active endocytosis. The paper could contribute to the understanding of the formulation factors that condition the efficacy and safety of lipid-based delivery systems for nucleic acid delivery. However, with this aim in mind authors should implement changes in the manuscript before the publication, according to the comments included in the attached file.

Author Response

Dear reviewer,

We thank for your detailed und very helpful comments and suggestions. They definitely made our work more valuable. In the following, we will address each of your comments individually, describe our changes in the manuscript, and discuss all open questions you raised.  All your aspects strongly improved our manuscript. All changes have also been incorporated in tracking mode into the revised version of our work.

Reviewer 1:

1 We have now clarified this aspect in the manuscript. For in vivo applications we additionally made very clear, that used fusogenic liposomes were explicitly produced (increased concentration to reduce necessary transfer volumes) for this work and provided by the company.

  1. We have adapted the material method part according to your helpful comments. Thank you very much for your comments. In addition, we have now also included the ethics statement.
  2. You are absolutely right that a consistent use of these terms depending on the investigation method is important to ultimately clarify the differences. We have now kept the term mRNA transfer in the qRT based analyses and replaced it by expression for the microscopy or flow cytometer analyses.
  3. Thank you for your very helpful comment. Characteristic features of fusion liposomes (Complex size and surface charge - doi: 10.3390/ijms21062244) and the characteristic differences between endocytotic and fusogenic liposomes (Changing the way of entrance - DOI: 10.1166/jbn.2019.2663 ) have already been published in previous publications. . To clarify this aspect we have reviewed the discussion section and indicated relevant publications or briefly described the properties of used fusogenic liposomes to better characterize the results shown here.

5. You are raising a very important question and we cannot answer this aspect at the moment. Based on previous studies (Complex size and surface charge - doi: 10.3390/ijms21062244) ) for fusogenic liposomes we can at least assume an almost 100% complexation of mRNA for complexation parameters used here. To mention this aspect, we have now included a short comment to encapsulation properties in the discussion section.

  1. We totally agree that transfection times, number of transfected cells and also amount of transferred mRNA molecules can have a major impact on inflammatory responses. To minimize these mainly protocol driven effects we have adapted the protocol for endosomal uptake, as indicated in Figure 1. While the standard lipofection protocol (SP) is based on long-term incubation and results in a strong inflammatory response, our adaptation for reduced uptake (RU) now allows better comparability of both systems. Now, difference in incubation times are reduced to result in comparable amounts of transferred RNA (qRT-PCR, indicated in fig. 1f) and similar protein amounts (GFP, Fig. 1c, d and e). We have checked the manuscript once again to better clarify the reason for this adaptation.
  2. You are absolutely right that a beta-MCD control incubation is essential. This is why we have done this accordingly in the respective experiment. Unfortunately, we did not describe this clear enough in the results section and in the figure legend. We have now clarified this aspect in the manuscript.
  3. Adapted in the manuscript.
  4. Adapted in the manuscript. We have added an additional information in the figure legend of fig. 4.
  5. A) The commercial composition of fusogenic liposomes contains a fluorescent dye for microscopic control of fusion. We therefore did not add any additional fluorescent compound to the product but just followed that dye after illumination. To clarify this aspect, we added a short note to the material and methods section. B) Thanks a lot for this helpful comment. We are now shortly mentioning this aspect in the discussion section. .
  6. All abbreviations are now explicitly explained upon first use in the manuscript.
  7. We have now added the conclusion section to the manuscript.

Reviewer 2 Report

see attached letter

Author Response

Dear reviewer,

We thank for your detailed und very helpful comments and suggestions. They definitely made our work more valuable. In the following, we will address each of your comments individually, describe our changes in the manuscript, and discuss all open questions you raised.  All your aspects strongly improved our manuscript. All changes have also been incorporated in tracking mode into the revised version of our work.

Reviewer 2:

Thank you for your numerous and very helpful comments. First, we would like to address your general comments. (1) The characteristic properties of fusogenic liposomes have already been described in detail in previous publications. However, you are absolutely right, that we needed to point out this fact more explicitly, wich is now done throughout the manuscript. (2) As you correctly noted, we have already gained a lot of experience with this comparison of systems and have also published other papers. The important difference and crucial advancement here is that we focused on the adjusted transfer amount of mRNA in the in vitro studies and based on these adjustments we were able to characterize the exact differences of given transfer route with respect to inflammatory responses of the cells as well as their effect on cytokines. In addition, such modifications allowed us to translate our protocols and analysis to an in vivo system and obtained comparable results to those shown here in the corresponding in vitro models.

  1. We have now included all requested information and adapted Table 1 accordingly.
  2. We apologize that this point was not sufficiently well described. For the in vitro studies, we used the established and commercially available fusogenic transfection system (Fuse-It-mRNA). For the in vivo studies, however, we neededa fusogenic transfection system of higher concentration for volume reasons. This system was provided by beniag as producing company. Since this in vivo system is not commercially available we have named the buffer formulation and described it here accordingly
  3. The reviewer is definitely right. All information about fusogenic liposome composition, complex charge and size have been published already and we are sorry of obviously not having this mentioned this well enough. All underlying references are now clearly indicated in the manuscript.

4.We have specified the amounts of transferred mRNA in the material and methods section as well in all figure legends.

  1. First of all, error bars represent SD of 3 independent experiments with in total 9 analyzed neuronal networks, which is now also indicated in the figure legend. Furthermore, you make a very interesting point here. We can currently only speculate about the high standard deviation after fusion. While neuronal connectivity and therefore function is fully blocked after lipofection, fusion does have a much more biocompatible effect on neuronal functionality. Since neuronal isolations and subsequent functional network formation can vary greatly, we believe that variations are mainly caused by different sample to sample sensitivity. Nevertheless, the data clearly proof high biocompatibility for fusion-dependent RNA transfer.

  1. The reviewer is certainly right that further information regarding underlying recognition receptors would be helpful to further characterize the differences between endocytosis and fusion-driven transfer. However, at this moment and based on the changes in cytokine mRNA expression after fusion and lipofection analyzed here, we only want to provide a first evidence that corresponding recognition receptors are activated with different intensity. Further experiment certainly need to done but, unfortunately, will be out of the scope of this manuscript. ?
  2. We have shortened and streamlined the discussion
  3. to 10. We are sorry of not having made clear enough the value of experiments performed here. To overcome this impression we included underlying papers describing exact characterization of fusogenic liposomes as basis of the commercial product Fuse-It-mRNA. Secondly, our data provide first evidence of low inflammatory responses upon direct RNA transfer into the cytosol. This high biocompatibility was not only characterized in classical cell culture systems but also at the level of functional neuronal networks and in animal experiments on zebrafish embryonic as well as adult level. Based on various adaptations throughout the manuscript we hope to have clarified the significance of our data.

Reviewer 3 Report

In this paper, the authors used fusogenic liposomes to establish highly efficient and fully biocompatible transfer of RNA molecules for in vitro as well as in vivo applications. The authors demonstrated that the fusogenic liposomes can bypass the endosomal uptake pathway, which has a lower immune response. This research application solved the problem of how to efficiently and biocompatibly transfer nucleic acid into animal cells. This study is significant. It may provide some new insights for mRNA delivery. However, some conclusions of this work have not been fully supported, and more data need to be supplemented.

Major comments

(1)    Although IL-6 or TNF-α was major inflammatory cytokines induced by foreign nucleic acid molecules, authors also should consider other kinds of inflammatory cytokines to verify experimental results. The authors would be better off adding 1-2 more classic inflammatory cytokines that represent different pathways.

(2)    In this paper, authors mainly detected inflammatory cytokines by RT-qPCR. However, the most accurate way to detect the expression of inflammatory cytokines is to identify the amount of protein expressed. For example, for the conclusion of line 308-310, the detection results of protein expression level were more accurate. I suggest authors should add the results of the protein expression of inflammatory cytokines.

(3)    It has been examined the inhibitory effect of MβCD on pro-inflammatory cytokine production in other systems (DOI: https://doi.org/10.1111/j.1365-201X.2007.01688.x). Therefore, we suggested authors should consider other methods of inhibition of endocytosis to verify the results that the reduced influence of endosomal uptake mechanisms for fusion-based RNA transfer allows low cytokine responses, such as hyperosmolar sucrose and K depletion.

(4)    In this section “Endosome-independent transfer mechanisms are highly efficient without detectable teratological effects in early embryogenesis”, why authors didn’t detect the expression of TNF-α, IL-6 or other inflammatory cytokines to verify the fusion-based RNA transfer allows low cytokine responses.

Minor comments

(1)    The green and red colors can be dimmed in statistics figures, such as fig 1e, fig 1f, fig 2c and so on.

(2)    Line 509-511, “fusogenic liposomes always showed significantly lower immunogenic effects, both in vitro and in vivo, due to significantly lower cytokine expression”. In fact, in this paper, the research on the effect of low levels of inflammation is limited. The above statement is not accurate.

Author Response

Dear Reviewer,

We thank for your detailed und very helpful comments and suggestions. They definitely made our work more valuable. In the following, we will address each of your comments individually, describe our changes in the manuscript, and discuss all open questions you raised.  All your aspects strongly improved our manuscript. All changes have also been incorporated in tracking mode into the revised version of our work. Below please find our point by point response.

Major:

(1) You raise a very important point here, which we also had in mind upon planning our experiments. For this manuscript and the in vitro/in vivo comparison, we decided to focus on PRR recognition of nucleic acids in endosomes. We expected this signaling cascade to most responsive to transfection reagents in order to have a most reliable and sensitive read-out platform. Ultimately, PRR recognition results in IL-6 and TNF-alpha cytokine expression. For clarification we also added additional information in the introduction section. ( Since we really could detect significant changes and different expression patterns for both cytokines after fusion and lipofection, PRR are certainly important, although not necessarily the only players. The exact interplay between PRRs and the analysis of further inflammatory cytokines in response to fusogenic and endocytic lipoplexes remains a fascinating question and is something we will strive for in further studies.

(2) We fully agree that measurement of cytokines on protein level is certainly more direct. However, by quantifying cytokine mRNAs by qRT-PCR, we aimed a detection of early changes in signal transduction and accordingly tried to identify immediate cellular processes as response to assumed PRR recognition of nucleic acids. Our data will certainly benefit from further analysis on protein detection level for various signal cascade players on in vitro as well as in vivo level, which will then be summarized in an independent manuscript.

(3) Thanks for this important comment. Exactly because of the study you mentioned, we had also performed experiments using chlorpromazine instead of beta-MCD to block endocytosis. However, this chemical compound strongly affected cell viability on its own and let to similar, however less reliable and reproducible results.

(4) The suggested experiment would certainly be of high interest. However, we were able to perform just a limited number of in vivo experiments on zebrafish embryos. For those embryos we subsequently focused on morphological development and staining rather than protein expression. By this we hope now to provide initial results on the influence on cellular developmental biology. The focus here for zebrafish embryo analyses was therefore not on the immune response and cytokine analysis that we performed in adult animals instead, but rather on more profound possible negative influences of the reagents on development. We tried to clarify our experimental focus in the results section.

Minor Comments:

  • Personally, we believe that using colors should only be considered if such labeling helps to understand presented data in a better way. For indicated statistics figures we are using two Y-axes for best data comparison. We tried to present figures in dimmed color or in grey scale. However, also due to sometimes closely spaced data points, readability was clearly impaired. We therefore would like to stick to the colors and color intensity used.
  • You are certainly right that we oversell our results at that point. We have therefore changed the text in the discussion section accordingly. The following statement now is purely based on our experiments. “Reduced expression of inflammatory cytokines, such as TNF-a and IL6, after treatment with fusogenic liposomes in vitro and in vivo may suggest a reduced immune response after treatment with this formulation”.

Round 2

Reviewer 1 Report

It is welcomed that the authors of the manuscript entitled “Smuggling on the nanoscale-Fusogenic lipsosomes enable efficient RNA-transfer with negligible immune response in vitro and in vivo” have revised properly the text, according to my comments. I just suggest a few last minor revisions to take into account before publication:

1.       I thank the inclusion of ethics statement regarding the isolation of freshly primary cells from rats, but authors have included it in the section that refers to In vivo experiments with zebrafish. Please, include the ethics statement in the section where the isolation of cells is described.

2.       Title of table 2: Replaces Assessed an treated by Assessed and treated.

3.       Table 2: delete the letter b in this table (up-left cell)

4.       Discussion section: thanks for the inclusion in this part of the manuscript of a more detailed description of fusogenic liposomes. However, authors should check the grammar and spelling of the following sentence:

“A precise characterization of the functional composition and phase transition of FLs was described Kolasinac et al. could mRNA FLs characteristics as complex size and surface charge was described by Hoffmann et al. and here we demonstrated that FLS transfer twice as much mRNA within 10 min as classical lipoplexes within 8 hours.

Moreover, authors must include the reference numbers of the papers published by Kolasinac et al. and Hoffman et al. that are mentioned in that sentence.

5.       Conclusions section has been properly included in the manuscript, but authors must check the first sentence: Transfection reagents for mRNA molecules are based on mainly based on two different transfer mechanisms into animal cells.

Author Response

  1. I thank the inclusion of ethics statement regarding the isolation of freshly primary cells from rats, but authors have included it in the section that refers to In vivo experiments with zebrafish. Please, include the ethics statement in the section where the isolation of cells is described.

Thanks for this input. We have the statement now transferred to the indicated section.

  1. Title of table 2: Replaces Assessed an treated by Assessed and

We have changed this.

  1. Table 2: delete the letter b in this table (up-left cell)

Also changed

  1. Discussion section: thanks for the inclusion in this part of the manuscript of a more detailed description of fusogenic liposomes. However, authors should check the grammar and spelling of the following sentence:

“A precise characterization of the functional composition and phase transition of FLs was described Kolasinac et al. could mRNA FLs characteristics as complex size and surface charge was described by Hoffmann et al. and here we demonstrated that FLS transfer twice as much mRNA within 10 min as classical lipoplexes within 8 hours.

Thanks for this comment. We have changed the wording and grammar accordingly.

Moreover, authors must include the reference numbers of the papers published by Kolasinac et al. and Hoffman et al. that are mentioned in that sentence.

Is now included.

  1. Conclusions section has been properly included in the manuscript, but authors must check the first sentence: Transfection reagents for mRNA molecules are based on mainly based on two different transfer mechanisms into animal cells.

Thanks for this very careful reading. We have adapted the sentence.

Reviewer 2 Report

Can the concentrations be provided in ug/mL as well *(rather than surface area).

Author Response

Can the concentrations be provided in ug/mL as well *(rather than surface area).

You are absolutely right. We have now changed the concentration as suggested throughout the whole manuscript.